# Role of Intraparotid and Neck Lymph Node Metastasis in Primary Parotid Cancer Surgery: A Population-Based Analysis

**DOI:** 10.3390/cancers14122822

**Published:** 2022-06-07

**Authors:** Mussab Kouka, Benjamin Koehler, Jens Buentzel, Holger Kaftan, Daniel Boeger, Andreas H. Mueller, Andrea Wittig, Stefan Schultze-Mosgau, Thomas Ernst, Peter Schlattmann, Orlando Guntinas-Lichius

**Affiliations:** 1Department of Otorhinolaryngology, Jena University Hospital, 07747 Jena, Germany; mussab.kouka@med.uni-jena.de (M.K.); bennykhlr@gmail.com (B.K.); 2Department of Otorhinolaryngology, Suedharzklinikum Nordhausen, 99734 Nordhausen, Germany; jens.buentzel@shk-ndh.de; 3Department of Otorhinolaryngology, Helios-Klinikum Erfurt, 99089 Erfurt, Germany; holger.kaftan@helios-gesundheit.de; 4Department of Otorhinolaryngology, SRH Zentralklinikum Suhl, 98527 Suhl, Germany; daniel.boeger@srh.de; 5Department of Otorhinolaryngology, SRH Wald-Klinikum Gera, 07548 Gera, Germany; andreas.mueller@srh.de; 6Department of Radiotherapy and Radiation Oncology, Jena University Hospital, 07743 Jena, Germany; andrea.wittig-sauerwein@med.uni-jena.de; 7Department of Oromaxillofacial Surgery and Plastic Surgery, Jena University Hospital, 07747 Jena, Germany; stefan.schultze-mosgau@med.uni-jena.de; 8University Tumor Center, Jena University Hospital, 07747 Jena, Germany; thomas.ernst@med.uni-jena.de; 9Department of Medical Statistics, Computer Sciences and Data Sciences, Jena University Hospital, 07743 Jena, Germany; peter.schlattmann@med.uni-jena.de

**Keywords:** parotid neoplasm, nodal metastasis, tumor staging, parotid lymph-nodes, intraparotid node, prognosis, incidence

## Abstract

**Simple Summary:**

The prognostic role of intraparotid (PAR) and cervical lymph node (LN) metastasis on overall survival (OS) of primary parotid cancer is unclear. All 345 Thuringian patients with parotid cancer from 1996 to 2016 were included in a population-based study. OS was assessed in relation to the total number of removed PAR and cervical LN, number of positive intraparotid (PAR+), positive cervical LN, LN ratio, log odds of positive LN (LODDS), as well as including the PAR as LODDS-PAR. PAR was assessed in 42% of the patients (22% of these PAR+). T and N classification were not independent predictors of OS. When combining T with LODDS instead of N, higher T became a strong prognosticator, but not LODDS. When combining T classification with LODDS-PAR, both higher T classification and the classification with LODDS-PAR became independent predictors of worse OS. LODDS-PAR seems to be an optimal prognosticator for OS in primary parotid cancer.

**Abstract:**

This population-based study investigated the prognostic role of intraparotid (PAR) and cervical lymph node (LN) metastasis on overall survival (OS) of primary parotid cancer. All 345 patients (median age: 66 years; 43% female, 49% N+, 31% stage IV) of the Thuringian cancer registries with parotid cancer from 1996 to 2016 were included. OS was assessed in relation to the total number of removed PAR and cervical LN, number of positive intraparotid (PAR+), positive cervical LN, LN ratio, log odds of positive LN (LODDS), as well as including the PAR as LODDS-PAR. PAR was assessed in 42% of the patients (22% of these PAR+). T and N classification were not independent predictors of OS. When combining T with LODDS instead of N, higher T (T3/T4) became a prognosticator (hazard ratio (HR) = 2.588; CI = 1.329–5.040; *p* = 0.005) but not LODDS (*p* > 0.05). When combining T classification with LODDS-PAR, both higher T classification (HR = 2.256; CI = 1.288–3.950; *p* = 0.004) and the alternative classification with LODDS-PAR (≥median −1.11; HR 2.078; CI = 1.155–3.739; *p* = 0.015) became independent predictors of worse OS. LODDS-PAR was the only independent prognosticator out of the LN assessment for primary parotid cancer.

## 1. Introduction

The function of the TNM staging system is not only to describe tumor size, regional lymph nodes, and distant metastasis but staging classification should also reflect the prognosis of individual patients. The N classification describing the neck nodal involvement in parotid cancer is a strong predictor for overall survival (OS) of patients in univariate analyses, but the results in multivariate analyses are controversial [1,2,3,4,5]. Hence, it is not clear if the N classification in parotid cancer is an independent predictor for OS. In this light, it was investigated if other lymph node-related factors, such as lymph node ratio (LNR or lymph node density) and the ratio of positive nodes to the total number of nodes removed (TNOD), may be a more effective marker. LNR was an independent prognostic factor for OS in a hospital-based series of high-grade salivary gland cancers [6]. Recently, LNR (but not the N classification) remained the only independent predictor of overall survival in a multivariate analysis in another hospital-based series focused on parotid cancer [7]. A relatively new lymph node classification scheme on OS is the log odds of positive lymph nodes (LODDS) [8]. LODDS is defined as the logarithm of the ratio between positive and negative lymph nodes. LODDS is able to discriminate between patients without positive lymph nodes, few nodes or insufficient nodes retrieved. LODDS has been demonstrated to be a prognostic classification superior to others such as pN and LNR [8,9], but has not yet been tested in parotid cancer. Furthermore, the current TNM classification does not consider the involvement of intraparotid lymph nodes (PAR) [5]. Intraparotid lymph nodes can be the only involved lymph node station in a high proportion of patients with parotid cancer, i.e., these patients are classified as N0 [10,11]. Intraparotid metastasis (PAR+) seems to be an independent prognosticator for worse overall survival in parotid cancer [5,12].

As is the case for many other types of head and neck cancer, there is a lack of detailed population-based studies on parotid cancer regarding the role of lymph node metastasis. Most knowledge results from larger monocentric series and retrospective studies and, therefore, has methodological limitations [1,5,6,7]. Clinical cancer registries allow one to explore the oncological outcome from a population-based perspective but do not allow to evaluate lymph node assessment and locoregional metastasis in detail [3].

Therefore, the present retrospective observational cohort study combined a population-based analysis on cancer registry data from all Thuringian patients treated for primary parotid cancer between 1996 and 2016 with a hospital-based approach. We retrieved the charts of all the registered patients to evaluate intraparotid and cervical lymph node resection and metastasis in detail, allowing us to calculate LNR and LOODS without and with inclusion of the intraparotid lymph node data.

## 2. Materials and Methods

The Ethics Committee of the Jena University Hospital approved the register studies (IRB No. 3204-07/11). The Ethics Committee waived the requirement for informed consent of the patients because the study had a non-interventional retrospective design and all data were analyzed anonymously. All procedures of the study involving human participants were in accordance with the Declaration of Helsinki (1964) and its later amendments or comparable ethical standards.

### 2.1. Study Design and Inclusion Criteria

Data of the Thuringian cancer registry database were the source for this population-based cohort study. The population-based Thuringian cancer registry combines data from the five Thuringian cancer registers (Nordhausen, Gera, Suhl, Jena and Erfurt) covering all cancer cases of Thuringia. Thuringia is a federal state in Germany with a population of about two million people. The Thuringian cancer registry covers about 98% of all head and neck cancer patients in Thuringia [13,14]. All new patients with primary parotid cancer registered between 1996 and 2016 were included. All cases of parotid cancer were classified according to the International Classification of Disease for Oncology, third edition, first revision (ICD-O-3) [15]. The inclusion criteria were as follows: primary carcinoma of the parotid gland treated, and in case of squamous cell carcinoma, no hint for primary skin cancer with parotid metastasis. Treatment was defined to be the first course of cancer-specific therapy of the primary tumor. Subsequent treatment for recurrent disease was not included in this definition of treatment. Patients who were treated for other major or minor salivary gland tumors, lymphoma, skin cancer or metastasis in the parotid region were excluded. Duplicate records of patients were removed. Patients’ flowchart is presented in Appendix A.

The extent of the disease was classified by clinical stages (cTNM) if no surgery was performed and by pathological stages (pTNM) when surgery was part of the therapy regime. Staging was defined by the AJCC Cancer Staging Classification, 7th edition (2010). Since the T or N classification were not clearly defined in all cases, stage grouping was not possible for all cases.

### 2.2. Intraparotid and Cervical Lymph Node Classification

In addition to the data from the cancer registries, patients’ charts in the five Thuringian hospitals treating the patients were reviewed for demographic characteristics, patients’ history, and details of the surgical treatment. All operation and histopathological reports were reviewed. The extent of the parotid surgery and neck dissection was captured. The total number of removed intraparotid lymph nodes (PAR) and the total number of positive lymph nodes (PAR+) was counted. The total number of removed ipsilateral cervical lymph nodes (TNOD) and the total number of ipsilateral positive lymph nodes (PNOD) was assessed. The classical lymph node ratio (LNR) was determined as the quotient from the PNOD and TNOD [8]. In a similar way, the intraparotid lymph node ratio (PARR) was calculated as the quotient from PAR+ and PAR. The natural logarithm of the quotient of PNOD and the number of negative lymph nodes is called LODDS (= log odds of positive lymph nodes) and was calculated as follows: log ((PNOD + 0.5)/(TNOD − PNOD + 0.5)). The value 0.5 was added to both PNOD (=positive number of lymph nodes) and TNOD (=total number of lymph nodes) in order to avoid a numerical singularity [8]. In a similar way, PARLODDS was calculated as follows: log ((PAR+ + 0.5)/(PAR − PAR+ + 0.5)). Finally, all lymph node calculations were also performed for the sum of intraparotid and cervical lymph nodes, including the total number of all lymph nodes (TNOD-PAR), total number of all positive lymph nodes (PNOD-PAR), lymph node ratio based on all harvested lymph nodes (LNR-PAR), and also the natural logarithm of the quotient including all intraparotid and cervical lymph nodes (LODDS-PAR).

### 2.3. Statistics

Statistical analyses were performed using IBM SPSS version 25.0 statistical software for Windows (Chicago, IL, USA). Differences between subgroups for patients with intraparotid lymph node assessment and the other patients were compared with Pearson’s chi-square test for nominal data and Fisher’s exact test for ordinal data. Overall survival (OS) was calculated by the Kaplan–Meier method. Differences of OS were compared by the log-rank test. Multivariable analysis was performed using the Cox proportional hazards model to estimate the hazard ratio (HR) with 95% confidence interval (CI) for OS. Only parameters with significant influence on OS in the univariate analysis (Kaplan–Meier analysis and log-rank test) were included into multivariate Cox models. These significant factors were grouped into several models (patients’ and tumor characteristics, treatment characteristics, classical staging, alternative staging using the new lymph node classifications). For all statistical tests, significance was two-sided and set to *p* < 0.05. 

## 3. Results

### 3.1. Baseline, Tumor and Therapy Characteristics

A total of 345 patients with primary parotid cancer were included. Table 1 shows the patients’ baseline characteristics, therapy and histology characteristics in detail. The median age was 66 years (range: 12–86). The gender distribution was almost equal (42.6% female). The five most frequent tumor subtypes were adenocarcinoma (22.3%), squamous cell carcinoma (17.4%), acinic cell carcinoma (11.6%), mucoepidermoid carcinoma (10.4%), adenoid cystic carcinoma (10.4%). Total parotidectomy was the most frequent parotidectomy type (47.5%), followed by lateral parotidectomy (18.3%). About half of the patients (47.5%) received a neck dissection and/or radiotherapy (50.7%). Chemotherapy was applied in 10.4% of the patients. Table 2 summarizes the tumor staging data. T classification was nearly equally distributed. Half of the patients were classified N0 (48.7%). About one third of the patients were stage IV (31.0%).

In less than half of the patients (42%), intraparotid lymph nodes were assessed. A total of 22% of these patients (33 of 145 patients) had intraparotid lymph node metastasis. To analyze the selection bias, the three subgroups (intraparotid lymph nodes examined, not examined, no parotidectomy performed) were compared (web appendix: Appendix A). The group of patients with assessed intraparotid lymph nodes were younger, more frequently stage III/IV, had more often a total parotidectomy and a neck dissection than patients without intraparotid lymph node examination. The patients that were not receiving a parotidectomy were older, more often M+, and had a higher probability to receive radiotherapy and especially chemotherapy.

### 3.2. Intraparotid and Cervical Lymph Node Assessment

More data on the intraparotid and cervical lymph node assessment are presented in Table 3. A total of 164 patients received a neck dissection. Selective neck dissection (57.3% of 164 patients) was more frequently performed than radical-modified neck dissection (30.9% of 164 patients; extent unknown: 11.5%). The median number of PAR was 3 (range: 1 to 22). PAR+ varied from 0 to 6 positive lymph nodes. The ratio PARR varied between 0 and 1. Median PARLODDS was −0.70 (range: −1.65 to −1.11). Median TNOD was 10 (range: 0 to 71). PNOD varied between 0 and 39 positive neck lymph nodes. Median LODDS was −1.23 (range: −3.30 to −1.65). For the total of the intraparotid and cervical lymph nodes, median TNOD-PAR was 11 (range: 0 to 71). PNOD-PAR varied between 0 and 39. Median LODDS-PAR was −1.11 (range: −3.30 to −1.65).

### 3.3. Overall Survival

Median follow-up was 42.5 months (range: 0 to 287). Median follow-up of patients alive was 58.5 months (range: 0 to 287). A total of 14.5% of the patients had a tumor recurrence. A total of 40.6% patients died within the observation period. Median OS was 126.00 months (CI = 104.12 to 147.89). The univariate analysis on parameters associated with worse OS is presented in the web appendix, Appendix A. Next to older age, TNM stage, histology and many of the lymph node parameters including PAR+ had a significant impact on overall survival. Some of the parameters are shown in Figure 1. 

The significant factors were grouped for the multivariate analyses. Model 1 grouped several patient, tumor and treatment characteristics (Table 4). Older age (>median of 66 years) had a higher risk of worse overall survival (HR = 2.896; CI = 1.924–4.359; *p* < 0.001). Squamous cell carcinoma (HR = 2.925; CI = 1.298–6.591; *p* = 0.010) and salivary duct carcinoma (HR = 4.287; CI = 1.469–12.509; *p* = 0.008) had a worse prognosis compared to non-specified carcinoma. Patients who did not undergo parotidectomy had worse prognosis (HR = 3.120; CI = 1.502–6.483; *p* = 0.002). If a neck dissection was performed, the prognosis was better than without neck dissection (HR = 0.553; CI = 0.372–0.820; *p* = 0.003). When combing therapy strategy and staging in model 2 (Table 5), neck dissection remained a positive prognosticator (HR = 0.488; CI = 0.331–0.720; *p* < 0.001). Higher tumor stage (III/IV) was associated to lower OS (HR = 1.842; CI = 1.206–2.813; *p* = 0.005). Within the TNM staging (model 3; Table 5), the M classification was the dominant prognosticator (HR = 4.047; CI = 1.705–9.607; *p* = 0.002). Even when taking the M classification out of the model (model 4; Table 5), T and N classification did not become independent predictors (both *p* > 0.05). When combining T with LODDS instead of N classification (model 5, Table 5), higher T classification (T3/T4) become a negative prognosticator (HR = 2.588; CI = 1.329–5.040; *p* = 0.005) but not LODDS (*p* > 0.05). Finally, when combining the T classification with LODDS-PAR (model 6, Table 5), both higher T classification (HR = 2.256; CI = 1.288–3.950; *p* = 0.004) and the alternative classification of the intraparotid and cervical lymph node status (≥median −1.11; HR 2.078; CI = 1.155–3.739; *p* = 0.015) become independent predictors of worse OS.

## 4. Discussion

Population-based studies on parotid cancer are rare but important. Such analyses reflect the outcome in clinical routine data beyond selected hospital-based data from specialist centers. Population-based studies are typically based on cancer registry data [3,16,17]. Hence, they are limited to the items that are registered. Detailed lymph node examination information is often not given or is fragmentary [4,18]. Data on intraparotid lymph node examinations are not part of cancer registries. The present study is, to our knowledge, the first to present detailed data on intraparotid and cervical lymph node assessments in a population-based setting, in combination with classification by new lymph node classificatory systems.

A recent systematic review including studies until 2020 revealed a mean pooled prevalence of the PAR+ rate of 24% [12]. The number of PAR+ lymph nodes varied from 1 to 11 in these hospital-based studies, which fits to the present study. The 5-year recurrence-free survival rate based on Kaplan–Meier analysis varied from 83% to 88% in PAR− patients, compared to 36% to 54% in PAR+. Only for studies allowed an OS analysis, the average calculated HR for a risk of death in patients with PAR+ compared to PAR− was increased by 2.14 [12]. In a newer analysis on major salivary gland cancer, the PAR+ rate was 21% and PAR+ 5-year OS rate was 57.6%, compared to 79.4% in PAR− patients [5]. This is in the same range as shown for parotid cancer in the present study and underlines the important impact of PAR+ on OS.

Several studies, even population-based studies, have already investigated lymph node parameters beyond the standard N classification. A surveillance, epidemiology, and end results (SEER) database analysis on major salivary gland cancer from 1998 to 2014 showed that a lymph node ratio of >0.15 was a prognostic indicator of cancer-specific survival [4]. Even more important, LNR remained a robust independent prognosticator in the multivariate analysis in this SEER study. This was confirmed by another SEER analysis including data from 1998 to 2010 combined with Chinese data [18]. Recently, it was shown that including the largest diameter of the nodes in combination with the number also produces a much more robust prognosticator. Standard N classification might not remain an independent predictor in multivariate analysis [5]. In the present study, it became clear that it is even more robust, first to include the intraparotid lymph node status into the regional lymph node assessment, and second to use the LODDS including the intraparotid and cervical status instead of the simple ratio. LODDS analysis on parotid cancer was not yet performed before the present study. LODDS was proven to be an independent and superior predictor for overall survival in several head and neck cancer series, some of them also including salivary gland cancer cases. LODDS is a better estimator than LNR and PNOD, especially since only a few positive lymph nodes were yielded. In addition, by adding 0.5 to the denominator and numerator, LODDS can also evaluate the OS of patients without positive lymph nodes [8,9,19].

Population-based studies based on cancer registry data have limitations. Many cancer registries, such as the German registries, do not collect systematically detailed data of the histopathological reports on intraparotid and cervical lymph node examination. To overcome this limitation, we analyzed, in addition, all the patients’ charts; hence, we linked the patients’ chart data with cancer registry data. There is no standard for the histopathology reports. It remains unclear why intraparotid lymph nodes were assessed in part of the patients but not in others. Moreover, due to the high amount of cases of parotid squamous cell carcinoma, we cannot conclude that some of these patients did not have a primary parotid cancer but an undiagnosed/not reported metastatic skin cancer. The important message of the present study, that PAR and LODDS-PAR is a very important prognostic factor, is equally relevant for these patients. Furthermore, and due to the retrospective cancer register-based approach, not all of the included patients received a total parotidectomy, an intraparotid lymph node assessment, and/or a neck dissection. Therefore, we cannot rule out a selection bias. Therefore, the presented results have to be confirmed in a prospective clinical trial with a standardized and systematic analysis of all the intraparotid lymph nodes in a large series of total parotidectomy cases. We could show that patients who did not receive a parotidectomy were older, more often M+, and had a higher probability to receive radiotherapy and especially chemotherapy. This limits the generalizability of our results. The presented results seem to be most relevant, of course, for patients without distant metastasis receiving a curative approach.

Lombardi et al. proposed to include the intraparotid lymph node assessment into the N classification [5]. The recent systematic review [12] and the present study support this proposal. To substantiate this request, a prospective and standardized assessment of intraparotid lymph nodes after parotid cancer surgery and standardized neck dissection, for instance by an international registry project, would be an important next step.

## 5. Conclusions

A large population-based study on patients with primary parotid carcinoma confirmed that intraparotid lymph node metastasis is an indicator for worse overall survival. Furthermore, the inclusion of the intraparotid lymph node status into the alternative lymph node assessment with the log odds of positive lymph node (LODDS) formula led to a robust prognosticator of the lymph node status in the multivariate analysis. In contrast, and confirming data of several other studies, standard N classification was less accurate in multivariate analysis. We recommend that a standardized assessment of intraparotid lymph nodes after parotid cancer surgery is always performed. A prospective clinical trial is needed to verify the present results. Hereafter, it should be considered to explicitly include the assessment of the intraparotid lymph nodes into the N classification of the UICC tumor staging.

## Figures and Tables

**Figure 1 cancers-14-02822-f001:**
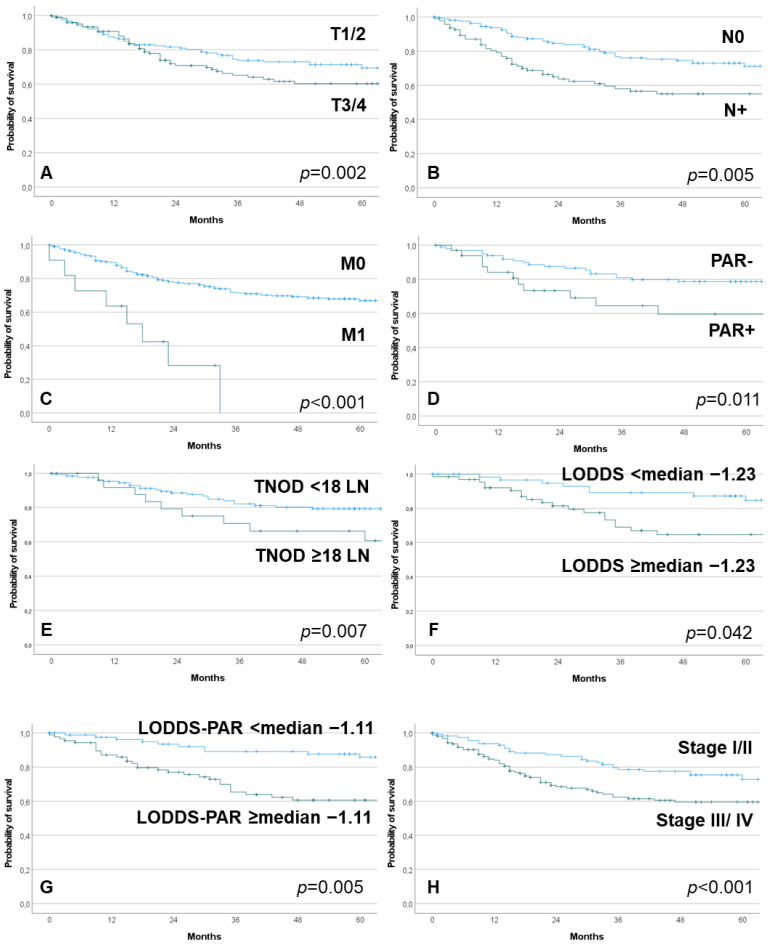
Kaplan–Meier curves on prognostic factors with influence on overall survival. (**A**): T classification; (**B**): N classification; (**C**): M classification; (**D**): intraparotid lymph nodes (PAR), PAR− = negative PAR; PAR+ = intraparotid metastasis; (**E**): total number of resected cervical lymph nodes (TNOD); (**F**): log odds of positive cervical lymph nodes (LODDS); (**G**): log odds of positive intraparotid and cervical lymph nodes (LODDS-PAR); (**H**): TNM stage.

**Table 1 cancers-14-02822-t001:** Baseline, therapy and histology characteristics.

Parameter	Absolute (*n*)	Relative (%)
All	345	100
Gender		
Female	147	42.6
Male	198	57.4
Parotidectomy		
Partial lateral	14	4.1
Lateral	63	18.3
Total	164	47.5
Radical	36	10.4
Not specified	41	11.9
No parotidectomy	27	7.8
Neck dissection		
No	181	52.5
Yes	164	47.5
Radiotherapy		
No	170	49.3
Yes	175	50.7
Chemotherapy/Biologicals		
No	309	89.6
Yes	36	10.4
Histology		
Adenocarcinoma	77	22.3
Squamous cell carcinoma	60	17.4
Acinic cell carcinoma	40	11.6
Mucoepidermoid carcinoma	36	10.4
Adenoid cystic carcinoma	36	10.4
Carcinoma not specified	24	7.0
Carcinoma ex-pleomorphic adenoma	24	7.0
Other rare carcinomas	17	4.9
Epithelial-myoepithelial carcinoma	14	4.1
Salivary duct carcinoma	11	3.2
Myoepithelial carcinoma	7	2.0
Undifferentiated carcinoma	6	1.7
Tumor recurrence		
No	295	85.5
Yes	50	14.5
Death		
No	205	59.4
Yes	140	40.6
**Parameter**	**Mean ± SD**	**Median, range**
Age (years)	63.9 ± 17.1	66, 12–86
Follow-up (months)	61.8 ± 58.7	42.5, 0–287
Follow-up of patients alive (months)	74.2 ± 60.6	58.5, 0–287

SD = standard deviation.

**Table 2 cancers-14-02822-t002:** Tumor staging.

Parameter	Absolute (*n*)	Relative (%)
All	345	100
TNM classification		
T1	75	21.7
T2	83	24.1
T3	70	20.3
T4	55	15.9
TX	61	17.7
N0	168	48.7
N1	24	7.0
N2	68	19.7
N3	2	0.6
NX	83	24.1
M0	309	89.6
M1	11	3.2
MX	25	7.2
AJCC stage		
I	62	18.0
II	53	15.4
III	50	14.5
IV	107	31.0
Unstaged	73	21.2
Summary stage		
Localized	167	48.4
Regional	89	25.8
Distal	11	3.2
Unstaged	78	22.6
Intraparotid lymph nodes		
No	173	50.1
PAR−	112	32.5
PAR+	33	9.6
No parotidectomy	27	7.8

PAR− = intraparotid lymph nodes without metastasis; PAR+ = intraparotid lymph node metastasis.

**Table 3 cancers-14-02822-t003:** Cervical lymph node harvest characteristics.

Parameter	Absolute (*n*)	Relative (%)
All	345	100
Neck dissection side		
Ipsilateral	161	46.7
Bilateral	3	0.9
No neck dissection	181	52.5
Neck dissection type ipsilateral		
Radical-modified/radical	51	14.8
Selective	94	27.2
Extent unknown	19	5.5
No neck dissection	181	52.5
Intraparotid lymph nodes examined		
No	173	50.1
Yes	145	42.0
No parotidectomy	27	7.8
**Parameter**	**Mean ± SD**	**Median, range**
Intraparotid lymph nodes (PAR)	3.65 ± 3.04	3, 1–22
Intraparotid lymph node metastasis (PAR+)	0.51 ± 1.10	0, 0–6
Intraparotid lymph node ratio (PARR)	0.16 ± 0.33	0, 0–1
Log odds of positive lymph nodes, parotid, (PARLODDS)	−0.56 ± 0.56	−0.70, −1.65–−1.11
Number of resected neck lymph nodes, ipsilateral (TNOD)	12.19 ± 10.61	10, 0–71
Positive neck lymph nodes, ipsilateral (PNOD)	1.82 ± 4.91	0, 0–39
Neck lymph node ratio, ipsilateral (LNR)	0.13 ± 0.26	0, 0–1
Log odds of positive lymph nodes, neck, ipsilateral (LODDS)	−1.08 ± 0.66	−1.23, −3.30–−1.65
Number of resected of neck and parotid lymph nodes, ipsilateral (TNOD-PAR)	13.01 ± 11.13	11, 0–71
Positive neck and parotid lymph nodes, ipsilateral (PNOD-PAR)	1.75 ± 4.52	0, 0–39
Neck and parotid Lymph node ratio, ipsilateral (LNR-PAR)	0.13 ± 0.26	0, 0–1
Log odds of positive lymph nodes, neck and parotid, ipsilateral (LODDS-PAR)	−0.94 ± 0.72	−1.11, −3.30–−1.65

**Table 4 cancers-14-02822-t004:** Cox regression: independent factors associated with worse overall survival.

Parameter	HR	95% CI Lower	95% CI Upper	*p*
**Model 1: Patient, tumor and treatment characteristics**				
Age, median: 66 years				
<Median	1	Reference	-	-
>Median	2.896	1.924	4.359	**<0.001**
Histology				
Carcinoma not specified	1	Reference	-	-
Adenocarcinoma	1.648	0.713	3.807	0.243
Squamous cell carcinoma	2.925	1.298	6.591	**0.010**
Acinic cell carcinoma	1.108	0.408	3.005	0.841
Mucoepidermoid carcinoma	1.058	0.362	3.089	0.918
Adenoid cystic carcinoma	0.829	0.311	2.207	0.707
Carcinoma ex-pleomorphic adenoma	2.643	0.982	7.112	0.054
Other rare carcinomas	2.315	0.741	7.237	0.149
Epithelial-myoepithelial carcinoma	0.655	0.188	2.278	0.506
Salivary duct carcinoma	4.287	1.469	12.509	**0.008**
Myoepithelial carcinoma	3.140	0.786	12.535	0.105
Undifferentiated carcinoma	2.338	0.647	8.444	0.195
Type of parotidectomy				
Lateral	1	Reference	-	-
Partial lateral	1.120	0.438	2.860	0.813
Total	1.896	1.092	3.291	**0.023**
Radical	1.615	0.773	3.377	0.203
Not specified	0.963	0.452	2.050	0.922
No parotidectomy	3.120	1.502	6.483	**0.002**
Neck dissection				
No	1	Reference		
Yes	0.553	0.372	0.820	**0.003**
Radiotherapy				
No	1	Reference	-	-
Yes	1.022	0.694	1.506	0.911
Chemotherapy/Biologicals				
No	1	Reference	-	-
Yes	1.380	0.805	2.364	0.241

HR = hazard ratio; CI = confidence interval. Significant *p*-values (*p* < 0.05) in bold.

**Table 5 cancers-14-02822-t005:** Cox regression *: independent factors associated with worse overall survival.

Parameter	HR	95% CI Lower	95% CI Upper	*p*
**Model 2: Treatment and staging characteristics**				
Neck dissection				
No	1	Reference	-	-
Yes	0.488	0.331	0.720	**<0.001**
Radiotherapy				
No	1	Reference		
Yes	1.094	0.721	1.661	0.673
Chemotherapy/Biologicals				
No	1	Reference		
Yes	1.603	0.967	2.658	0.067
AJCC Stage				
I/II	1	Reference	-	-
III/IV	1.842	1.206	2.813	**0.005**
**Model 3: Traditional staging I**				
T classification				
T1/2	1	Reference	-	-
T3/4	1.457	0.949	2.237	0.085
N classification				
N0	1	Reference	-	-
N+	1.405	0.915	2.157	0.121
M classification				
M0	1	Reference	-	-
M1	4.047	1.705	9.607	**0.002**
**Model 4: Traditional staging II**				
T classification				
T1/2	1	Reference	-	-
T3/4	1.492	0.981	2.267	0.061
N classification				
N0	1	Reference	-	-
N+	1.517	0.997	2.309	0.052
**Model 5: Alternative staging I**				
T classification				
T1/2	1	Reference	-	-
T3/4	2.588	1.329	5.040	**0.005**
LODDS, median −1.23				
<Median	1	Reference	-	-
≥Median	1.809	0.921	3.551	0.085
**Model 6: Alternative staging II**				
T classification				
T1/2	1	Reference	-	-
T3/4	2.256	1.288	3.950	**0.004**
LODDS-PAR, median −1.11				
<Median	1	Reference	-	-
≥Median	2.078	1.155	3.739	**0.015**

HR = hazard ratio; CI = confidence interval; LODDS = log odds of positive lymph nodes, neck, ipsilateral; LODDS-PAR = log odds of positive lymph nodes, neck and parotid, ipsilateral. * The Cox regression models with same parameters but for continuous numbers/ordinal scales are presented in the web appendix: Appendix A. * In addition, we also performed Cox regression models including all significant parameters of Table 4 into the models of Table 5 in the web appendix: Appendix A and web appendix: Appendix A. Significant *p*-values (*p* < 0.05) in bold.

## Data Availability

The datasets used during the current study are available from the corresponding author upon reasonable request.

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
