# Peer review of "Role of Intraparotid and Neck Lymph Node Metastasis in Primary Parotid Cancer Surgery: A Population-Based Analysis"

_cancers, 2022, doi:10.3390/cancers14122822_

Round 1

Reviewer 1 Report

Prof. Guntinas-Lichius and his coworkers evaluated the role of intraparotid and cervical neck nodes in primary parotid cancer. 

First of all, I absolutely agree that the value of intraparotid lymph nodes is definitely underestimated in the current staging system and therefore this work is highly relevant. However, whether complex ratios such as PAR, PAR+, PARR, TNOD, PNOD, LNR, PARLODDS, LODDS, LODDS-PAR are really worthy remains another issue. Personally, I do believe that these ratios make the story more complex than necessary.

This study has definitely its value, but carries a lot of flaws that need to be addressed first.

1) Study Cohort: 

  • In total, 345 patients with primary parotid cancer have been evaluated including also 60 patients with SCC. As we all know, primary parotid SCC basically do not exist and so these cases typically present intraparotid SCC metastasis. Therefore, these patients need to be excluded.
  • The 27 cases without parotidectomy should be also excluded as they do not provide data regarding intraparotid lymph nodes
  • Neck Dissection was applied in "only" 47.5% of cases, which means in turn that almost half of the patient cohort was not eligible for lymph node ratios. 

2) Statistics

  • Did the authors assess also other outcome parameters than OS? E.g. CSS
  • How did the authors select variables for multivariable cox-regression analyses? 
  • Why did the authors do not test for normally distributed variables?

3) Data / Analysis

  • how did the authors explain the difference between 11 patients with M1 disease but only 7 cases staged as distal disease - Table 2).
  • This is maybe the most important issue - Did the authors check for any significant associations between variables, such as age, gender, T-classification, histology,... and evaluated ratios? Or in other words, how do the authors know that their ratios and not T-classification is the most relevant factor? Maybe an additional Table with metric data of Table 3. 
  • Univariate survival analysis applying log-rank tests revealed that the most significant factors for poor OS were T-classification, M-classification,  tumor stage and N-classification followed by the LODDS-PAR ratio. Importantly, these analyses do not support the need for any ratios. Why did the authors use TNOD<18 as cut-off?
  • Regarding Table 4 - is this also a multivariable analysis? Its not clear for me. And if so, why did the authors choose these variables? RT, ChT, type of parotidectomy and ND seem to be no ideal candidates. 
  • Similarly comments to Table 5 - this model type analysis is quite unusual and not really comprehensible. A multivariable analysis should include all significant factors that have been previously identified. Otherwise the importance of even highly significant results is questionable or at least overdrawn.  

Altogether, I still believe that the study has its value, but actually it seems to me that the authors tried a bit too hard to find significances to confirm the meaning of their ratios. I would really appreciate if analyses would be done again after data clearing. 

Author Response

Reviewer #1

Prof. Guntinas-Lichius and his coworkers evaluated the role of intraparotid and cervical neck nodes in primary parotid cancer. First of all, I absolutely agree that the value of intraparotid lymph nodes is definitely underestimated in the current staging system and therefore this work is highly relevant. However, whether complex ratios such as PAR, PAR+, PARR, TNOD, PNOD, LNR, PARLODDS, LODDS, LODDS-PAR are really worthy remains another issue. Personally, I do believe that these ratios make the story more complex than necessary.

This study has definitely its value, but carries a lot of flaws that need to be addressed first.

1) Study Cohort: 

1.1. In total, 345 patients with primary parotid cancer have been evaluated including also 60 patients with SCC. As we all know, primary parotid SCC basically do not exist and so these cases typically present intraparotid SCC metastasis. Therefore, these patients need to be excluded.

Answer 1.1: In the methods, we write: “The inclusion criteria were: primary carcinoma of the parotid gland treated, and in case of squamous cell carcinoma, no hint for primary skin cancer with parotid metastasis.” We excluded 206 (!) patients with other primaries, including all cases with skin cancer history and squamous cell carcinoma metastasis. The number of included patients were parotid squamous cell cancer without skin cancer accounted for 17.4% of the sample. Depending on the definition, the incidence of primary squamous cell varies from 0.1% to 10% (Ying YL, Johnson JT, Myers EN. Squamous cell carcinoma of the parotid gland. Head Neck. 2006 Jul;28(7):626-32. doi: 10.1002/hed.20360. PMID: 16475198.; Fang Q, Wu J, Liu F. Oncologic outcome and potential prognostic factors in primary squamous cell carcinoma of the parotid gland. BMC Cancer. 2019 Jul 31;19(1):752. doi: 10.1186/s12885-019-5969-6. PMID: 31366378; PMCID: PMC6669973.). Although this suggests that, may be, at least half of the 17.4% were theoretically carcinoma of unknown primary (undetected/not reported skin cancer) these patients were treated like all other patients with primary parotid cancer. A subanalysis of this 17.6% patients (unpublished; not shown in the manuscript) shows that LODDS-PAR is also a powerful predictor in this subgroup. The present study was a population-based study on real-world data, hence showing how these patients are treated in real life and daily practice. The important message that PAR and LODDS-PAR is as important in this subgroup as for any other subtype of parotid cancer, hence to recommend total parotidectomy and neck dissection also in this subgroup.

To take up this important issue raised by reviewer #1, we revised the text as follows in the Discussion. In the section in limitations (page 10), we added: “Moreover, due to the high amount of cases of parotid squamous cell carcinoma, we cannot exclude that some of these patients had not a primary parotid cancer but an undiagnosed/not reported metastatic skin cancer. The important message of the present study that PAR and LODDS-PAR is a very important prognostic factor, is equally relevant for these patients.”

1.2. The 27 cases without parotidectomy should be also excluded as they do not provide data regarding intraparotid lymph nodes

Answer 1.2: What differs the present study from many others, it the population-based perspective. As we commented already in answer 1.1, this approach reflect the clinical routine in an representative way. Instead of excluding these patients, which would mean to include a severe selection bias, we deliberately included these patients and performed additional analyses between the patients with and without parotidectomy (see supplement tables). This approach revealed important results. To address the typical limitations of such a cancer registry based study we added in the Discussion, in the paragraph on the limitations on page 11: “Furthermore, and due to the retrospective cancer register-based approach, not all included patients received a total parotidectomy, an intraparotid lymph node assessment, and/or a neck dissection. Therefore, we cannot rule out a selection bias. We could show that patients not receiving a parotidectomy were older, more often M+, and had a higher probability to receive radiotherapy and especially chemotherapy. This limits the generalizability of our results. The presented results seem to be most relevant, of course, for patients without distant metastasis receiving a curative approach”

1.3. Neck Dissection was applied in "only" 47.5% of cases, which means in turn that almost half of the patient cohort was not eligible for lymph node ratios. 

Answer 1.3: See also answer 1.1 and 1.2, because this query refers to the same topic: selection of patients in a population-based study. There is a lack of lack of conceptual clarity in the international and also the German clinical guidelines concerning neck dissection in case of parotid cancer. The results of the presents study are showing the consequence in clinical routine: a high variable behavior in performing or not performing a neck dissection. The results is comparable to other population-based or even many hospital-based studies. The limitation, also concerning the neck dissection, were also ready mentioned in Answer 1.2 and are addressed now in the Discussion. Furthermore, we added the importance of the neck dissection in the last paragraph of the Discussion. We write now on page 11: “To substantiate this request, a prospective and standardized assessment of intraparotid lymph nodes after parotid cancer surgery and standardized neck dissection, for instance by an international registry project, would be an important next step”

2) Statistics

1.4. Did the authors assess also other outcome parameters than OS? E.g. CSS

Answer 1.4: No, we restricted the analysis to OS. The analyses is already very extensive, including supplementary material.

1.5. How did the authors select variables for multivariable cox-regression analyses? 

Answer 1.5: We added information in the Methods/Statistics part as follows on page 4. “Only parameters with significant influence on OS in univariate analysis (Kaplan-Meier analysis and log-rank test) were included into multivariate Cox models. These significant factors were grouped into several models (patients’ and tumor characteristics, treatment characteristics, classical staging, alternative staging using the new lymph node classifications).”

1.6. Why did the authors do not test for normally distributed variables?

Answer 1.6: The primary outcome parameter of this study was overall survival using Kaplan-Meier statistics and Cox regression. There was no need to perform an analysis for normal distribution before including parameters in these statistics. May be we do not understand, why reviewer #1 wants us to perform such statistics. For the main aim, these statistics were not needed.

3) Data / Analysis

1.7. how did the authors explain the difference between 11 patients with M1 disease but only 7 cases staged as distal disease - Table 2).

Answer 1.7: Many thanks for the hint. There was a error in table 2! We corrected the data. There were N=11 patients with M1. We corrected the presentation of the summary stage, parameter known to be presented in U.S. American cancer registries like the SEER or the NCDB databases. Theses changes had no influence on the statistical calculations.

1.8. This is maybe the most important issue - Did the authors check for any significant associations between variables, such as age, gender, T-classification, histology,... and evaluated ratios? Or in other words, how do the authors know that their ratios and not T-classification is the most relevant factor? Maybe an additional Table with metric data of Table 3. 

Answer 1.8: The primary and only relevant outcome parameter of the present study was overall survival. As shown Table S2, all the parameters listed in query 1.8 were analyzed regarding their influence on OS. All parameter with significance were then included into thze multivariate analyses. It was the task to the multivariate analyses to detect or rule out dependencies of the mentioned factors. Using the classification staging (models 3, 4 in Table 5), T only showed a trend to be important as independent factor (p <0.1, but not p<0.05). T became a relevant factor when combining it with alternative lymph nodes classificators (model 5, 6 in Table 5). Hence, yes, T is important, but this was only evident when using the newer alternative lymph node classificators.

1.9. Univariate survival analysis applying log-rank tests revealed that the most significant factors for poor OS were T-classification, M-classification,  tumor stage and N-classification followed by the LODDS-PAR ratio. Importantly, these analyses do not support the need for any ratios. Why did the authors use TNOD<18 as cut-off?

Answer 1.9: The p-values in Table S2 do not allow a ranking concerning the role of each parameter as independent risk factor. This has to be proven by multivariate analyses. In the present study, this was performed by Cox regression, as overall survival was the primary outcome criterion. Furthermore, as explained in detail in Answer 1.11, many of the factors analyzed in the univariate survival analyses, are per se, associated to each other. Examples: UICC stage to T, N, M; N to TNOD, PNOD, LODDS, LOODS-PAR.

TNOD 18: Actually, as shown in Table S2, we analyzed several cut-offs. As explained already, and in detail in Answer 1.11, it does not make sense to include several cut-offs (for instance, TNOD<>18 and TNOD<>16, both significant) in the same model, as these two dichotomic variables would be directly dependent to each other. So, we could have chosen both parameter in different models, but showed only data for TNOD<>18 as a lymph node yield of 18 lymph nodes has to be shown in several studies as an important cut-off for improved survival, see for instance: de Kort WWB, Maas SLN, Van Es RJJ, Willems SM. Prognostic value of the nodal yield in head and neck squamous cell carcinoma: A systematic review. Head Neck. 2019 Aug;41(8):2801-2810. doi: 10.1002/hed.25764. Epub 2019 Apr 10. PMID: 30969454; PMCID: PMC6767522.

1.10. Regarding Table 4 - is this also a multivariable analysis? Its not clear for me. And if so, why did the authors choose these variables? RT, ChT, type of parotidectomy and ND seem to be no ideal candidates. 

Answer 1.10: Yes, Table 4 and Table 5 are showing all Cox regression models, hence, all multivariate analyses. In the legend of Table 4, the words “Cox regression” were missing. These words are added now. Please see answer 1.5. As it is usual, we included all parameters showing influence on survival in different Cox models. We grouped them into reasonable groups excluding to combine parameters that are per se dependent to each other.

1.11. Similarly comments to Table 5 - this model type analysis is quite unusual and not really comprehensible. A multivariable analysis should include all significant factors that have been previously identified. Otherwise the importance of even highly significant results is questionable or at least overdrawn.  

Answer 1.11: A multivariable analysis can (!) include all significant factors that have been previously identified. There is no “must” as explained above. First, a researcher is completely free to combine factors as she/he wants in Cox models (Schneider A, Hommel G, Blettner M. Linear regression analysis: part 14 of a series on evaluation of scientific publications. Dtsch Arztebl Int. 2010 Nov;107(44):776-82. doi: 10.3238/arztebl.2010.0776. Epub 2010 Nov 5. PMID: 21116397; PMCID: PMC2992018.; Zwiener I, Blettner M, Hommel G. Survival analysis: part 15 of a series on evaluation of scientific publications. Dtsch Arztebl Int. 2011 Mar;108(10):163-9. doi: 10.3238/arztebl.2010.0163. Epub 2011 Mar 11. PMID: 21475574; PMCID: PMC3071962). Second, if an factor is “highly” significant, which is often the case for M+, it can be that such a factor is so dominating, that the relevance of other factors gets lost. In such a situation, it can also make sense, and is allowed in Cox modelling, to calculate a model with M and other excluding M.

In addition, as it is the case in the present study, the univariate analyses revealed parameters with significant association to worse overall survival that are dependent to each other. For instance, UICC staging was significant, but also T, N, and M. It does not make sense and is even against basic statistical laws to combine for instance T and UICC staging in the same Cox model, as the UICC staging includes T, hence the UICC staging is per se dependent on T. So, T and UICC can never be independent factors. The same hold trues for the parameter in focus of this study: TNOD, PNOD, LODDS and so on. All of them are associated to N. It does not make sense to combine them in the same model out of statistical reasons, but also in regard of the main question of the manuscript. The aim was to analyze the alternative lymph node classification systems. Therefore, these systems were alternatively included into different Cox models.

1.12. Altogether, I still believe that the study has its value, but actually it seems to me that the authors tried a bit too hard to find significances to confirm the meaning of their ratios. I would really appreciate if analyses would be done again after data clearing. 

Answer 1.12: See the answers of the other queries. We followed a clear rationale and statistical rules. It is this the intention of the Cox regression to analyze the association between selected parameters. As it is usual, we only combined parameters with significance in univariate analysis. In addition, of course, we did not combine parameters dependent per se to each other. The main aim in the modelling was to compare the classical staging with alternative stagings.

Reviewer 2 Report

Kouka et al., aimed to present a retrospective observational cohort study combined a population-based analysis on cancer registry data from all Thuringian patients treated for primary parotid cancer between 1996 and 2016 with a hospital-based approach. Moreover, they went back to the charts of all registered patients to evaluate intraparotid and cervical lymph node resection and metastasis in detail allowing to calculate LNR and LOODS, without and with inclusion of the intraparotid lymph node data.

The study covers some issues that have been overlooked in other similar topics. The structure of the manuscript appears adequate and well divided in the sections. Moreover, the study is easy to follow, but few issues should be improved. Some of the comments that would improve the overall quality of the study are:

a-) Authors must pay attention to the technical terms acronyms they used in the text.

b-) English language needs to be revised.

c-) Conclusion Section: This paragraph required a general revision to eliminate redundant sentences and to add some "take-home message".

Author Response

Thank you very much for the detailed and helpful reviews. We answer here all queries/comments.

Reviewer #2

Kouka et al., aimed to present a retrospective observational cohort study combined a population-based analysis on cancer registry data from all Thuringian patients treated for primary parotid cancer between 1996 and 2016 with a hospital-based approach. Moreover, they went back to the charts of all registered patients to evaluate intraparotid and cervical lymph node resection and metastasis in detail allowing to calculate LNR and LOODS, without and with inclusion of the intraparotid lymph node data. The study covers some issues that have been overlooked in other similar topics. The structure of the manuscript appears adequate and well divided in the sections. Moreover, the study is easy to follow, but few issues should be improved. Some of the comments that would improve the overall quality of the study are:

2.1 Authors must pay attention to the technical terms acronyms they used in the text.

Answer 2.1: We checked the text again. We consequently used always the same abbreviations. We used the abbreviation mostly used in studies analyzing the role of lymph node harvest and cervical lymph node metastasis. Hence, PNOD, TNOD, LNR, and LODDS are the abbreviations normally used in all these study. In addition, PAR is a typical abbreviation of for parotid lymph nodes. It was therefore consistent, and this is new, because it was not done before, also to combine the abbreviation PAR with the other abbreviations. In this first draft of the paper, we still wrote out the terms when combining PAR aspects with the lymph node algorithms. In this form, the text was not readable. In the present form with all these abbreviations, the text is much better readable.

2.2 English language needs to be revised.

Answer 2.2: The original text was already revised by a native speaker. The new version of the manuscript was revised again. To our knowledge, minor editing is also done by the Cancers editorial team.

2.3 Conclusion Section: This paragraph required a general revision to eliminate redundant sentences and to add some "take-home message".

Answer 2.3: The main take-home message of the conclusions is expressed in the last sentence. This was revised now more in the style of a take-home message: “We recommend to always perform a standardized assessment of intraparotid lymph nodes after parotid cancer surgery. The assessment of the intraparotid lymph nodes should be explicitly included into the N classification of the UICC tumor staging.”

Orlando Guntinas-Lichius

for all authors

Jena, 02-April-2022

Reviewer 3 Report

In my opinion the topic of this article is one of interes. It is may be a little bit difficult to pursue the text because extensive use of the shortcuts of the terms. 

I think may be of interest to discus about the difficulties in intraparotid lymph node metastasis from squamous cell carcinoma (from a cutaneous cancer) extended into parotid tissue. In some cases, is difficult to make the difference with origin of the squamous neoplasm in parotid  gland tissue 

Author Response

Thank you very much for the detailed and helpful reviews. We answer here all queries/comments.

Reviewer #3

3.1. In my opinion the topic of this article is one of interest. It is may be a little bit difficult to pursue the text because extensive use of the shortcuts of the terms.

Answer 3.1: Same as query 2.1 of reviewer #2.  We consequently used always the same abbreviations. We used the abbreviation mostly used in studies analyzing the role of lymph node harvest and cervical lymph node metastasis. Hence, PNOD, TNOD, LNR, and LODDS are the abbreviations normally used in all these study. In addition, PAR is a typical abbreviation of for parotid lymph nodes. It was therefore consistent, and this is new, because it was not done before, also to combine the abbreviation PAR with the other abbreviations. In this first draft of the paper, we still wrote out the terms when combining PAR aspects with the lymph node algorithms. In this form, the text was not readable. In the present form with all these abbreviations, the text is much better readable.

3.2. I think may be of interest to discuss about the difficulties in intraparotid lymph node metastasis from squamous cell carcinoma (from a cutaneous cancer) extended into parotid tissue. In some cases, is difficult to make the difference with origin of the squamous neoplasm in parotid  gland tissue 

Answer 3.2: See also query 1.1 of reviewer #1. To take into account this important query we revised the Discussion: In the section in limitations (page 10), we added: “Moreover, due to the high amount of cases of parotid squamous cell carcinoma, we cannot exclude that some of these patients had not a primary parotid cancer but an undiagnosed/not reported metastatic skin cancer. The important message of the present study that PAR and LODDS-PAR is a very important prognostic factor, is equally relevant for these patients.”

Orlando Guntinas-Lichius

for all authors

Jena, 02-April-2022

Round 2

Reviewer 1 Report

I thank the authors for their responses to the questions I raised.

Unfortunately, I am also not yet convinced by the data or statistical analyses and the conclusions drawn from them. I would recommend to revise the whole manuscript, exclude unclear or questionable data and to especially involve a statistician in the analyses.  

Although I choose to reject, I still believe in the value of the topic and that the manuscript will benefit from significant corrections. 
